# FGF/FGFR Signaling in Hepatocellular Carcinoma: From Carcinogenesis to Recent Therapeutic Intervention

**DOI:** 10.3390/cancers13061360

**Published:** 2021-03-17

**Authors:** Yijun Wang, Danfei Liu, Tongyue Zhang, Limin Xia

**Affiliations:** Department of Gastroenterology, Institute of Liver and Gastrointestinal Diseases, Hubei Key Laboratory of Hepato-Pancreato-Biliary Diseases, Tongji Hospital of Tongji Medical College, Huazhong University of Science and Technology, Wuhan 430030, China; M202076163@hust.edu.cn (Y.W.); D201981720@hust.edu.cn (D.L.); M201976049@hust.edu.cn (T.Z.)

**Keywords:** FGF, FGFR, HCC, targeted therapies

## Abstract

**Simple Summary:**

As the most common primary liver cancer, HCC is a tricky cancer resistant to systemic therapies. The fibroblast growth factor family and its receptors are gaining more and more attention in various cancers. Noticing an explosion in the number of studies about aberrant FGF/FGFR signaling in HCC being studied, we were encouraged to summarize them. This review discusses how FGF/FGFR signaling influences HCC development and its implications in HCC prediction and target treatment, and combination treatment.

**Abstract:**

Hepatocellular carcinoma (HCC) is the most common type of primary liver cancer, ranking third in cancer deaths worldwide. Over the last decade, several studies have emphasized the development of tyrosine kinase inhibitors (TKIs) to target the aberrant pathways in HCC. However, the outcomes are far from satisfactory due to the increasing resistance and adverse effects. The family of fibroblast growth factor (FGF) and its receptors (FGFR) are involved in various biological processes, including embryogenesis, morphogenesis, wound repair, and cell growth. The aberrant FGF/FGFR signaling is also observed in multiple cancers, including HCC. Anti-FGF/FGFR provides delightful benefits for cancer patients, especially those with FGF signaling alteration. More and more multi-kinase inhibitors targeting FGF signaling, pan-FGFR inhibitors, and selective FGFR inhibitors are now under preclinical and clinical investigation. This review summarizes the aberrant FGF/FGFR signaling in HCC initiating, development and treatment status, and provide new insights into the treatment of HCC.

## 1. Introduction

As one of the most common malignant tumors of the liver, hepatocellular carcinoma (HCC) is a major public health issue worldwide with increasing morbidity and cancer-related mortality but limited intervention options and low curative rates. Chronic liver disease associated with hepatitis B virus (HBV) or hepatitis C virus (HCV) is the most common etiology of HCC, especially in developing areas. Approximately 80% of HCC patients worldwide have HBV or HCV infection. Non-alcoholic fatty liver disease (NAFLD) and diabetes are the primary and increasing risk factors for HCC in developed countries. The consumption of aflatoxin B1, cigarettes, and alcoholic substances are also associated with HCC [1,2,3]. Although there are various kinds of interventions for HCC, only a few early-diagnosed patients can receive potential curative therapies through surgical resection, transarterial chemoembolization (TACE), or ablation. Even worse, HCC is a highly insidious cancer, meaning that HCC is often detected at intermediate or even advanced stages and therefore miss the optimal treatment window. In these cases, systemic pharmacological treatment is the best therapy but can only provide modest benefits [3,4]. Despite the many surveillance protocols, the overall survival of HCC is still unsatisfactory. Agent acquired resistance, the high recurrence rate after surgical resection, and the lack of biomarkers for HCC identification highlight the need for investigators to further elucidate the molecular pathology of HCC and provide more alternative therapeutic options.

The family of fibroblast growth factors (FGFs) consist of eighteen ligands and four homologous factors. FGFs bind with corresponding FGF receptors (FGFR1~4) and are involved in various actions, including early embryogenesis, angiogenesis, wound repair, metabolism, and many other physiological processes in adults [5,6]. The downstream effects of FGF/FGFR, such as regulating cell proliferation, differentiation, and survival, indicate that this axis is a potential target in the pathogenesis of multiple tumors. Several cancers, such as breast cancer, gastric cancer, endometrial cancer, bladder cancer, myeloma, and HCC, can be induced when FGF/FGFR signaling is abnormal [7,8,9,10,11]. Thus far, interventions targeting FGF/FGFR have provided modest benefits for patients, and some of these treatments have already been approved in the clinic. Furthermore, FGF/FGFRs show value as biomarkers for patient identification, which is essential for detecting the interpatient heterogeneity of HCC. All of the observations above indicate that targeting FGF/FGFR signaling is a promising therapeutic option for HCC patients. In this review, we summarize the recent discoveries relating to FGF/FGFR signaling in HCC. We discuss the roles of FGF/FGFR signaling in HCC initiation and development, and the treatment status and provide new insights into the treatment of HCC.

## 2. The FGF/FGFR System

### 2.1. FGF, FGFR, and Co-Factor

FGF ligands are categorized into five paracrine subfamilies (FGF1, FGF2; FGF4, FGF5, FGF6; FGF3, FGF7, FGF10, FGF22; FGF8, FGF17, FGF18), one endocrine subfamily (FGF15/19, FGF21, FGF23) and several homologous factors (FGF11-14) (Table 1). FGF11-14 share substantial sequence homology with other FGFs, but they have neither recognizable secretory signal peptides nor the ability to be secreted from cells; thus, their bioactivity is intracellular and they cannot initiate any FGFR signaling pathways. Therefore, these molecules are not discussed in this review [12]. Mouse FGF15 is orthologous to human FGF19, sharing 51% amino acid identity [13].

FGFRs are divided into three domains as transmembrane receptors: three extracellular Ig-like loop domains (termed Ig I, Ig II, and Ig III), two intracellular tyrosine kinase domains (termed TK1-2), and a single transmembrane helix linking the extra and intra components. Ig-loop II and Ig-loop III are essential for signal transduction because they specifically recognize ligands and cofactors [14]. In contrast, a conserved stretch of 7–8 acidic residue sequences, called the acid box, is involved in receptor autoinhibition and signaling inhibition along with Ig I. Four FGFRs share high sequence similarities, especially FGFR1-3. Alternative splicing of FGFR is often observed in Ig-loop III of FGFR1-3, generating two isoforms generally named IIIb and IIIc (Figure 1). The FGFR IIIb isoform tends to be expressed in epithelial cells. While the FGFR IIIc isoform is more likely to be expressed on mesenchymal cells, the transformation from IIIb to IIIc is associated with epithelial-mesenchymal transition (EMT) [15]. FGFR4 lacks an alternative splicing exon and has no isoform. Given the core role of ligand recognition in Ig-loop III, the isoforms alter the ligand-receptor binding spectrum [15].

To signal, FGFs bind to FGFR in the presence of cofactors: heparin sulfate (HS) proteoglycans (HSPGs), or Klotho protein. The affinity for different cofactors determines the endocrine, paracrine, and autocrine actions. HSPGs exist widely in the extracellular matrix. Nearly all FGFs have HS binding domains but differ in terms of affinities. On the one hand, HSPGs potentiate signaling transmission for morphogens and growth factors by functioning as accessory receptors. On the other hand, FGFs with strong association associations are tethered nearby and act in a paracrine or autocrine manner. Contrarily, FGFs with a low affinity for HSPG (FGF19, FGF21, and FGF23) enter the circulation and exert their functions through endocrine actions [16]. Without HS, those endocrine FGFs utilize klotho proteins to serve as coreceptors, which confer stability and preferential binding of endocrine FGFs to their respective FGFRs. There are two homologs of klotho protein that promote different FGF signaling pathway: β klotho (also named KLB) and α klotho (also called KL) [16]. Multiple lines of evidence demonstrate that KLB is essential for the coactivation of FGF19 and FGF21 signaling, while FGF23 tends to bind with KL to initiate the pleiotropic cellular function [17,18]. Unlike the HS in the extracellular matrix, Klotho protein has obvious tissue specificity, ultimately determining the various roles of these endocrine FGFs in different tissues. KLB is widely expressed in the liver and fat and preferentially binds with FGFR1c and FGFR4. Therefore, signals from FGFR4 and its ligand FGF9 are among the major FGF signaling pathways in HCC [19].

### 2.2. FGF/FGFR Downstream Signaling

To signal, secreted FGFs bind to heparin sulfate and heparin sulfate binding sites (HBS) of FGFR, forming signal-transducing dimers. Such conformational changes enable the autophosphorylation and activation of intracellular tyrosine kinases [14]. Activated FGFRs then phosphorylate docking proteins such as FGFR substrate 2 (FRS2) and FGFR substrate 3 (FRS3) [20]. These effectors can function as scaffolds and finally activate four intracellular pathways: mitogen-activated protein kinase (MAPK); phosphatidylinositol 3-kinase (PI3-kinase), phospholipase Cγ (PLCγ), and signal transducer and activator of transcription (STAT). These factors play important roles in proliferation, metastasis, angiogenesis, and agent-acquired resistance in the initiation and development of HCC (Figure 1).

## 3. Deregulation of FGF/FGFR in HCC

### 3.1. FGF2

FGF2 is expressed in HCC cells and is barely detected in nonparenchymal cells or noncancerous liver tissue [21]. By targeting the IIIc isotype of FGFR1 as its primary receptor, FGF2 is involved in the development of HCC as a potent mitogen. This molecule stimulates DNA synthesis of hepatoma cell lines and promotes tumor growth. It also enhances the synthesis of the plasminogen activator, promoting tumor cell invasion [21]. Additionally, FGF2 is a potent mitogen for endothelial cells (ECs), vascular smooth muscle cells (VSMCs), and mural cells, participating in HCC angiogenesis [22]. FGF2 and FGF19, seem to be involved in maintaining cancer stem-like cells (CSCs) with CD^44^High/CD^133^High cell membrane markers, which play a central role in the tumor microenvironment, potentially enhancing HCC tumorigenesis, metastasis, and anticancer drug resistance [23]. Furthermore, FGF2 is reported to be involved in the immune regulation of HCC. FGF2 enhances the sensitivity of NK cells to tumor cells by upregulating the expression of membrane-bound major histocompatibility complex class I-related chain A (MICA) and suppressing the expression of human leukocyte antigen (HLA) class I, which are activating molecule and inhibitory molecules of NK cells, respectively [24]. From this point of view, FGF2 may partly contribute to the elimination of innate immunity in HCC cells. GAL-F2 is a specific monoclonal antibody for FGF2. This molecular inhibits the proliferation and migration of HCC cell lines and blocks angiogenic signals in the mouse model, corroborating the role of FGF2 in tumor growth and vascularization [25].

### 3.2. FGF8 Subfamily

Although widely expressed during embryonic development, the expression of FGF8 subfamilies is mainly restricted to hormonal cancers such as prostate cancer and breast cancer during adulthood [26]. However, the abnormal signaling of FGF8 subfamilies (FGF8, FGF17 and FGF18) have also been detected in many other cancers [8,26,27]. By a paracrine and autocrine mechanism, FGF8, FGF17, and FGF18 participate in the development of HCC. In vitro, FGF8 promotes the proliferation of HCC cell lines. Moreover, FGF8 increases the expression of EGFR through transcriptionally upregulating Yes-associated protein 1 (YAP1), which contributes to resistance to EGFR inhibitors [28]. FGF17 is a potent mitogen for prostate cancer. This molecule can be induced by FGF8, indicating its potential mediating role in FGF8 function [29]. FGF18 impaires apoptosis while enhancing cell proliferation, motility, and invasion in HCC [30]. Knockout of FGF18 dramatically reduces the malignant phenotype of cells [31]. According to research conducted by Gauglhofer et al. [26], the levels of at least one FGF8 subfamily member and/or one FGFR are upregulated in 82% of HCC cases. Additionally, the co-upregulation of the levels of at least one FGF and one FGFR is detected in approximately one-third of these tumor. The researchers also noted that the Wnt pathway and hypoxia-inducible transcription factors might be two possible mechanisms that regulated FGF8, FGF17 and FGF18 overexpression in HCC [26]. In addition to contributing to the tumorigenic characteristics of tumor cells, FGF18 subfamilies are also involved in the angiogenesis of HCC, which will be discussed in the subsequent chapters.

### 3.3. FGF9

FGF9 expression is often co-upregulated with FGFR3 IIIb/IIIc expression in HCC patients. Furthermore, FGF9 is the most dominant ligand for FGFR3 IIIb/IIIc in HCC [32]. FGF9 is secreted mainly by HSCs in normal and cirrhotic livers and acts by the paracrine mechanism to stimulate tumor cells. However, this molecule has an autocrine effect in HCC since tumor cells are the primary source at this stage. This result is inconsistent with other research, which noted out that activated HSCs/myofibroblasts are the primary sources of FGF9, while no FGF9 expression was detected in either HCC cells and hepatocytes [33]. However, both findings emphasize the tumorigenic role of FGF9 in HCC. FGF9 exerts its tumorigenic role by specifically binding to FGFR3, facilitating the proliferation and migration of HCC cell lines and promoting new blood and lymphatic capillary formation [32]. Moreover, FGF9 is involved in sorafenib resistance, indicating that FGF9 may be a promising target of HCC. Pan-inhibitors of FGFR or siRNA targeting FGFR3 could block the oncogenic properties of FGF9 in HCC [33].

### 3.4. FGF19 Subfamily

This family is closely associated with multiple metabolic regulatory processes, including insulin resistance, fatty acid oxidation, bile acid, triglycerides, and glycogen [6]. Many of these metabolic processes are found in the liver and have close relationships with liver physiology and pathology status. FGF19, FGF21, and FGF23 can facilitate HCC in both metabolism-dependent and metabolism-independent pathways. This chapter will discuss the metabolism-independent roles, and the metabolic effects in HCC development will be discussed in the following section.

In hepatocytes, FGF19 mainly targets FGFR4 as its receptor, with KLB stabilizing their integration and interaction. FGF19 is primarily expressed and secreted from ileum villus epithelial cells and gallbladder epithelial cells in adults. FGF19 can also be secreted by cells from pathological liver tissue, such as cholestatic noncirrhotic and cirrhotic livers and livers from individuals with alcoholic hepatitis and HCC [34]. Additionally, FGF19, FGFR4, and KLB appear to increase with hepatic pathology, from steatosis to steatohepatitis, cirrhosis, and finally HCC [35]. Transgenic mice with ectopically expressed FGF19 exhibited preneoplastic changes, including constitutive hepatocellular proliferation and AFP expression. At 10–12 months, an average of 53% of the mice had developed locally invasive HCCs. In p53^−/−^ mice, all FGF19 transfected mice died within the 100-day observation period, while none of those in the control groups died [36]. These results support the direct effect of FGF19 on HCC initiation [37]. With the coaction of KLB, FGFR4 initiates a growing number of intracellular signaling pathways to target tumor cells, promoting hepatoma cell proliferation and migration and inhibiting tumor cells apoptosis. FGF19 mediates cell escape from death by increasing the expression and phosphorylation of IL-6-induced STAT3, which is known to lead to compensatory proliferation in tumor cells [38,39]. Another newly discovered target gene of FGF19 in HCC is SOX18. SOX18 is an oncogene promoting the proliferation and metastasis of tumor cells in many cancers. FGF19/FGFR4 upregulates the expression of SOX18 through p-FRS2/p-GSK3β/β-catenin signaling. Interestingly, SOX18 is also a ligand for FGFR4, forming positive feedback among SOX18, FGF19, and FGFR4 in HCC development. BLU9931, which is a selective FGFR4 inhibitor, significantly inhibits the growth of SOX18-induced HCC metastasis [40].

However, FGF19 also exerts protective effects on the liver. Mitogenic FGF19 deficiency delays liver regrowth and impairs hepatocyte regeneration after chemical liver injury or partial hepatectomy in a mouse model. Similarly, the knockout of FGFR4 or siRNA application increases the susceptibility of the liver to CCL4 exposure. This phenomenon mechanism can be partly attributed to bile acid accumulation resulting from FGF19 deficiency. Additionally, the proliferative signals provided by FGF15 are also indispensable for the regeneration since a cholate-supplemented diet cannot compensate for the growth impairment in FGF15-null mice [41]. NF2/Merlin might control the shunting of pro-oncogenic and antioncogenic signaling of FGFR4. NF2/Merlin is an upstream regulator of the Hippo pathway and is activated by FGFR4 signaling to maintain various organ sizes. NF2/Merlin might act as a switch in FGFR4 signaling by activating ERK and attenuating Mst1/2-mediated signaling [42].

Unlike other FGFs in HCC, FGF21 expression is usually decreased in HCC and is believed to protect the liver. Multiple lines of evidence have shown that FGF21 maintains metabolic homeostasis and contributes to antifibrotic processes during the development of HCC [43,44]. FGF21 is reported to relieve acute or chronic inflammatory diseases by inhibiting the production of IL-17A, which has recently been proven to be associated with human hepatitis, fatty livers, and viral hepatitis-associated HCC [45]. Both rhFGF21 administration and blockage of IL-17A benefited the liver in terms of arresting progressive liver diseases [44,45].

### 3.5. Other FGFs in HCC

FGF5 has been discussed in many other cancers and has multiple roles in cancer development [46,47,48]. FGFR1 IIIc and FGFR2 IIIc are considered the preferential receptors for FGF5. In pancreatic cancer, FGF5 promotes pancreatic cancer cells growth through its binding to FGFR1 IIIc [47]. Fang et al. [49] noted that FGF5, as a downstream molecule of miR-188-5p, is involved in the proliferation, colony formation, cell migration, and invasion of SMMC7721 cells, promoting carcinogenesis in HCC by activating H-Ras—p-ERK signaling. However, no research has been conducted to identify the exact receptor of FGF5 in HCC and its biological functions in vivo. FGF7 has been reported to participate in the nucleotide excision repair (NER) pathway by upregulating ERCC1 expression via the FGFR2-ERK pathway [50]. ERCC1 has been identified as a critical rate-limiting enzyme during the NER process. The overexpression of ERCC1 reflects the higher activity of NER related to HCC resistance to platinum drugs [51].

While some FGFs may have an oncogenic role in other organs, their roles in HCC remain to be clarified. These FGFs include FGF6, FGF10, FGF20, and FGF22. FGF6 is strongly overexpressed in prostate cancer tissues compared with normal prostate tissues, stimulating the transformed of prostatic epithelial cells [52]. However, FGF6 poorly expressed in normal liver tissues and HCC, and it accumulates almost exclusively in the myogenic lineage [53]. FGF10 is involved in multiorgan development, and its knockout may cause severe dysmorphia [54]. Otherwise, FGF10 is thought to act as a mediator of androgen action, thus contributing to prostate cancer pathogenesis by facilitating epithelial proliferation [55]. FGF22 in the brain is reported to be associated with depression. To date, no research has been performed to illustrate the role of these FGFs in HCC.

### 3.6. FGFRs in HCC

FGFR3 and FGFR4 are the major FGFRs overexpressed in HCC, while upregulation of FGFR1 and FGFR2 expressing are rarely observed [56,57]. FGFR3 expression has been reported to be upregulated in 17 of 32 HCC cases. In five cases displaying upregulation of IIIb expression, eight cases showed upregulation of IIIc expression, and four cases showed both. The ligands of FGFR3 are different according to the variants. The ligands of the FGFR3-IIIb variants are mainly FGF1 and FGF9. In comparison, the FGFR3-IIIc variants have additional binding sites for FGF2, FGF4, FGF8, FGF17, FGF18, FGF19, FGF21, and FGF23. Although the two isoforms provide different FGFs docking sites and transmit different downstream signals, upregulation of the expression of both enhances hepatoma cells malignant phenotypes [57]. FGFR4 shows the highest expression in the liver compared with other major organs. Additionally, hepatocyte is the only cell type where FGFR4 is more dominant than all other FGFRs [58]. In HCC, FGFR4 expressions is upregulated in nearly half of HCC cases, along with a major increase in different FGF ligands, such as the FGF2 and FGF8 subfamilies [59]. Although all of these ligands can bind with FGFR4, signals from FGF19 seem to be exerted preferentially and cannot be replaced by any other ligands [59].

## 4. Aberrate FGF/FGFR Signaling

The FGF-FGFR pathway can be activated aberrantly in many ways. Gene mutation, amplification, and chromosome translocation all contribute to the overactivation of FGF/FGFR signaling. Many upstream molecules can stimulate FGF and thus transmit signals. Some molecules, such as miRNAs, can even function as ligands and directly interact with FGFR. The most common aberration caused by FGF ligands is gene amplification. Although FGFR mutation or amplification is quite common in many cancers, it is not a frequent event during hepatocarcinogenesis [57].

### 4.1. Mutation

Most of the FGFR3 mutations occur in exons 7, 10, and 15. These mutations are common in bladder and uterine cervical cancer [60,61]. However, they have not detected in HCC [57]. FGFR3^△7–9^, which is a mutant formed by directly linking exon 6 to exon 10, has been proven to promote the malignant phenotype of HCC. This mutation may change its ligand-binding domain, making it capable of autoactivation and self-phosphorylation [62]. FGFR3^△7–9^ significantly downregulates E-cadherin expression while upregulating Snail and MMP-9 expression, which may partly explain how FGFR3^△7–9^ enhances malignant phenotypes [62]. FGFR3^△7–9^ can also phosphorylate and degrade tumor suppressor ten-eleven translocation-2 (TET2) and its target gene PTEN, contributing to HCC cell proliferation [63].

### 4.2. Amplification

FGF3, FGF4, and FGF19 are located at the 11q13.3 amplicon, which is one of the most frequent amplification sites in HCC and other cancers. The FGF3/4 and FGF19 amplification frequencies are approximately 2–3% and 20%, respectively [36,64]. FGF19 is located near known oncogene CCND1 and is often observed to be coamplified with CCND1. The detection of TCGA liver hepatocellular carcinomas (LIHC) demonstrates that all of the genetic alterations of FGF19 exhibit amplification instead of gene mutation or deletion [65]. In human HCC oncogenic screening conducted by Sawey et al. [36], 14% of HCC samples were observed to harbor a focal amplicon (<10 Mb) containing FGF19 with a good RNA/DNA association. The amplification of FGF19 can explain approximately 6% of FGF19 overexpression in HCC. As a driver event, FGF19 amplification seems to provide a selective advantage to the evolution of HCC, meaning that patients with amplified FGF19 are more likely to succumb to HCC. Furthermore, amplified FGF19 is closely associated with lower 5-year OS (overall survival), along with a larger tumor size, multiple tumors, and microvascular invasion [66]. The amplification of FGF19 is also closely associated with HCC histological subtypes. Macrotrabecular-massive HCC (MTM-HCC) is a proliferative and highly aggressive phenotype of HCC that is genetically characterized by TP53 mutations and/or FGF19 amplification, accounting for 10–20% of all cases of HCC [67]. This subtype more frequent in patients infected with HBV and is clinically associated with early tumor recurrence and poor overall survival [68]. Gene expression profiling revealed that MTM-HCC is highly related to G3-subgrouped tumors, which have high chromosome instability rates and are show cell cycle/proliferation/DNA metabolism-related genes overexpression [67,68]. Whether these characteristics entirely or partially explains the FGF19 proliferative and metabolic properties of amplified tumors is a fascinating question. MTM-HCC is also characterized by angiogenic activation with angiopoietin 2 and vascular endothelial growth factor A (VEGFA) overexpression [68]. However, few articles have demonstrated the relationship between FGF19 and angiogenesis; that is, the two factors may just be mechanically correlated. Nevertheless, the correlations between molecular changes and disease phenotypes are promising for personalized medicine. Moreover, an investigator found that the amplification of FGF19 has a good relationship with the clinical response to sorafenib. Patients who showed a complete response to sorafenib harbor a higher frequency of FGF19 copy number variations than those without a complete response (*p* = 0.024, chi-squared test); that is, FGF19 amplification can also be biomarker to predict sorafenib response [69]. The amplification of FGF3 and FGF4 is considered to have less clinical importance. Although patients with amplified FGF3 and FGF4 are more likely to respond to sorafenib [64,70], the frequency of FGF3/FGF4 amplifications is relatively low, restricting its predictive value in the clinic. Combining FGF3/FGF4 amplifications with multiple lung metastases and other events to predict sorafenib responsiveness has been suggested [64]. Additionally, the amplification of FGF3 and FGF4 is barely associated with an increase in the corresponding mRNA or protein, thus showing a limited biological function during HCC development.

## 5. FGF/FGFR Signaling in Angiogenesis

HCC is a highly vascularized tumor and heavily relies on angiogenesis for tumor growth. Vascular endothelial growth factor A (VEGF-A), also known as VEGF, is one of the major factors contributing to new blood vessel formation. Thus far, most antiangiogenic therapies have revolved around anti-VEGF strategies. Sorafenib, which is an antiangiogenic agent targeting VEGF, was approved by the FDA in 2007 [71]. Although it has been proven to prolong median survival and the time to progression by nearly three months in patients with advanced HCC, this treatment shows no increase in survival rate when it is combined with other chemotherapies over the years [71]. Moreover, the mechanism by which sorafenib benefits patients or patient tolerance to treatment remains unclear [72]. Further elucidation of the proangiogenic factors underlying this process and new insights to assist with anti-VEGF therapy are needed.

The FGF1, FGF2, FGF4, and FGF8 subfamilies are the most frequently studied FGFs in the angiogenic processes of HCC. Among these factors, FGF2 is the best known and earliest characteristic factor. FGF2 mainly targets FGFR1 as its receptor to mediate angiogenic. FGF2 exerts various roles during multiple angiogenic stages [73]. As a potent mitogen for ECs, VSMCs, and mural cells, FGF2 directly triggers new vessel formation by promoting in their proliferation. FGF2 also synergizes with other angiogenic factors such as VEGF and platelet-derived growth factor-BB (PDGF-BB), to potentiate mutual angiogenic effects [22,74]. Multiple lines of evidence have demonstrated the synergistic effect between VEGF and FGF2. FGF2 upregulates VEGFA expression in various types of cells, including HCC tumor cells. VEGF mRNA increases 3.1-fold in cells with FGF2-overexpression in murine HCC [74]. The tumor growth induced by FGF2 overexpression can be significantly inhibited by the VEGFR2 monoclonal antibody, indicating VEGF’s role as a downstream mediator in FGF2-induced angiogenesis [74,75]. For the same reason, many anti-VEGF therapies confer resistance due to the compensating role of FGF2. Increasing research is now focusing on the dual blockage of these factors [76].

PDGF-BB is a potent stimulator of VSMCs, but not ECs. The main reason for this is the lack of PDGFR expression on ECs membrane. FGF2 is reported to transcriptionally upregulate the PDGFR on ECs, thus potentiating the PDGF-BB-induced migration of ECs and compensating for FGF2 merely as a mitogenic factor for ECs. Conversely, enhancing PDGF-BB upregulates the expression of FGFR1 on VSMCs, which promotes the proliferative effect of FGF2 on VSMCs. The interplay between these two factors increases the density and disorder of the newly-formed vessel, making it more consistent with tumor vessels’ properties [22]. Another example demonstrating the synergism of these angiogenic factors is the PDGF-BB-PDGFR pathway involved in pericyte adhesion. Pericytes are cells surrounding the monolayer of ECs, the affinity of which contributes to the integrity of the basement membrane. A deficiency in pericytes is responsible for aberrant microvascular formation. VEGF induces the secretion of PDGF-BB from ECs. Moreover, FGF-2 can upregulate PDGFR expression in mural cells. FGF2 and VEGF cooperate to enhance the mural cell attachment to ECs by increasing the PDGFR expression in mural cells and PDGF-BB secretion from ECs [77].

FGF8 subfamily members are also involved in angiogenesis. These factors promote the proliferation and tube-forming ability of human umbilical vein endothelial cells (HUVECs), exerting a direct role in HCC [26]. In HCC, FGF18 can be induced by Wnt/β-catenin and secreted from HCC cells and HUVECs. FGF18 binds to its receptor FGFR3 and activates downstream AKT and ERK signaling to mediate angiogenesis in HCC [78]. In addition, FGFR3 promotes the proliferation and tube formation of HUVECs through monocyte chemotactic protein 1 (MCP-1) [79]. In addition to modulating tumor cells and endothelial cell activity, FGF8 subfamily members can mediate the tumor microenvironment function by upregulating VEGF expression by stimulating myofibroblast DNA synthesis. In summary, aberrant FGF8/17/18 signaling participates in HCC angiogenesis in both a direct and indirect way. Notably, FGF18 is detected in liver sinus endothelial cells, indicating that it can stimulate endothelial cells through both autocrine and paracrine mechanisms [26,80].

FGF16 is also detected in human cardiovascular tissues [80]. HUVECs treated with recombinant FGF16 were shown to have enhanced migration without enhancement of proliferation. The same results were observed in the recombinant FGF18 treatment group. A reasonable interpretation of this is that FGF16 and FGF18 only activate MAPK briefly, rather than in a sustained way. The transient activation of MAPK by FGF16 and FGF18 is sufficient to enhance migration but not proliferation [80]. However, it is worth noting that there has been no evidence proving the expression of FGF16 in the liver to date.

## 6. FGF/FGFR Signaling in Metabolism

FGF19, FGF21, and FGF23 regulate systematic metabolic homeostasis of nutrients and minerals via their endocrine effects on multiple organs. FGF19 is a postprandial hormone and exerts similar insulin roles, such as stimulating glycogen and protein synthesis and inhibiting gluconeogenesis in the liver (Figure 2) [81]. FGF19 is also involved in the regulation of BA metabolism, which is closely associated with several liver diseases [19,65,82]. In contrast, FGF21 is a fasting-induced hepatokine and is partly comparable to glucagon, inducing a transformation to catabolic metabolism through binding to β Klotho-FGFR1c [16]. FGF23 is secreted from osteocytes as a response to phosphate intake [83]. FGF23 is involved in the metabolism of several minerals, including phosphate, sodium, calcium, vitamin D, and PTH [83,84,85]. These three metabolic FGF subfamilies regulate a wide range of metabolic procedures, and the disruption of homeostasis may result in various diseases. In this chapter, we will discuss metabolic FGF-associated liver damage and HCC.

### 6.1. FGF19

Bile acid is synthesized in the liver and secreted into the small intestine as a critical component of the bile, where it aids in the digestion of fat and lipid nutrients. However, excessive concentrations of bile acids can induce liver damage and even the malignant transformation of hepatocytes [86]. Additionally, increasing cellular cholesterol impairs lysosomal function and results in accumulation of autolysosomes, impairing hepatic autophagy and contributing to hepatocyte injury [87]; that is, the homeostasis of bile acid and cellular cholesterol is crucial to a healthy liver. The farnesoid X receptor (FXR) is the primary sensor of bile acid and inhibits bile acid synthetic metabolism to maintain BA at a biosafe concentration. In this process, FGF15/19 is the central effector in the FXR-induced feedback inhibitory loop of bile acid synthesis. After feeding, an increasing flux of bile acid enters the intestine, leading to the activation of FXR followed by an increase in FGF15/19. FGF15/19 inhibits cholesterol 7α-hydroxylase (CYP7A1), which is the key and speed-limited enzyme of bile acid metabolism, forming the inhibitory feedback loop of bile acid synthesis [80]. Apart from the regulation of CYP7A1, FGF19 also stimulates the relaxation and refiling of the gallbladder to regulate BA synthesis [88].

### 6.2. FGF21

Unlike other FGFs, FGF21 has no effect as a mitogen but exerts critical roles in regulating glucose and lipid metabolism. FGF21 mediates fatty acid oxidation and ketogenesis in response to starvation through a peroxisome proliferator-activated receptor (PPAR) α-dependent mechanism [89]. Although FGF21 is mainly produced by the liver, adipocytes are the principal targets of FGF21, where FGFR1c and βKlotho are both highly expressed [16]. FGF21 targets the white adipocytes to stimulate glucose uptake, ketogenesis, and lipolysis and engages in thermogenic regulation through targeting the brown adipocytes [90,91]. To expand FGF21 metabolic functions from the fat issue, FGF21 upregulates expression of adiponectin, which is an adipokine secreted by adipocytes, to systematically mediate pleiotropic metabolic benefits in the endocrine mode [91]. FGF21 protects liver function and attenuates hepatic pathology caused by various long-term or short-term nutritional challenges, such as a ketogenic diet and alcohol, an obesogenic diet, methionine, a choline-deficient diet, and a high fructose diet [89,92,93,94,95]. For example, when treated with a high-fat diet, FGF21 KO mice showed increasing liver pathology in 16 weeks and severe fibrosis or even HCC in 52 weeks, which was not observed in control WT mice [93].

## 7. FGF/FGFR and Tumor Metastasis

Metastasis is an important characteristic for distinguishing malignant tumors from benign tumors. It is the end phase of cancer progression and accounts for more than 90% of cancer mortality. The separation of malignant cells from the primary lesion is the first step of metastasis. In this process, these cells undergo a transition from an epithelial to a mesenchymal phenotype, namely, EMT, which enables them to disperse from the original focus. These free cells then infiltrate locally, break through the basement membrane, and finally enter the circulation as metastatic circulating tumor cells, finding suitable sites to resettle. FGF2 and FGF19 have been proven to be associated with the EMT process during HCC development. Signals transmitted from FGF19/FGFR4 can phosphorylate and activate GSK3β and promote β-catenin translocation into the nucleus, where it functions as a transcription factor, suppressing the expression of several key genes in the Wnt pathway [96]. More evidences have appeared recently. Small surface antigens (SHBs) are a component of the hepatitis B surface antigen, making it the most abundant HBV virus in the serum of HBV patients. Infecting SHBs induces endoplasmic reticulum stress, leading to the expression and autocrine activity of FGF19. Secreted FGF19 targets FGFR4 and mediates the EMT progression of HCC by upregulating the expression of several EMT-associated transcription factors, such as ZEB1, snail, slug, and twist, by activating JAK2/STAT3 signaling [97]. FGF2 is considered a critical mediator of EMT in numerous cancers, such as bladder cancers, breast cancer, and HCC [98,99,100]. When the HCC cell lines LH86 and Huh7 are treated with FGF2, both show changes associated with EMT, such as an increase in vimentin, fibronectin, and collagen I levels, and a decrease in E-cadherin levels [100]. FGF1 has also been reported to regulate the EMT of HCC. Both HepG2 and SMMC-7721 cells show EMT associated protein expression changes and enhanced invasion when treated with the FGF1 plasmid [101]. As a functional target of miR-188-5p, FGF5 promotes the migration and invasion of SMMC7721 cells by activating H-Ras-p-ERK, while the downregulation of FGF5 expression shows the opposite results [49]. FGF9 binds to FGFR3 IIIb or IIIc and enhances cell line migration by inducing gaps in monolayers of blood or lymphatic endothelium, similar to tumor invasion into the circulation. Furthermore, treatment with recombinant FGF9 significantly enhanced HCC migration [32]. Moreover, FGF9 can regulate the HSC cell lines’ migration by creating an inflammatory microenvironment in NAFLD mice, contributing to liver fibrosis and HCC development [35]. All these lines of evidence illustrate that multiple FGFs can have roles in HCC migration. The blockage of these pathways shows promise for tumor control.

## 8. FGF/FGFR and Drug Resistance

HCC is a cancer with high molecular heterogeneity cancer with plenty of potential oncogenic drivers but no clear addiction loops, making the acquired resistance of targeted tyrosine kinase a different problem. With an increasing number of preclinical trials supporting the therapeutic effects of FGF/FGFR in HCC, the role of FGF/FGFR in acquired resistance has been identified and cannot be ignored. For example, FGF8 induces resistance to EGFR inhibitors by upregulating the expression of yes-associated protein 1 (YAP1) and EGFR [28]. FGF9 may also participate in sorafenib resistance. Treatment of rFGF9 interferes with sorafenib’s effects on HCC cells via JNK pathway activation [33]. Genomic profiling has also revealed that the enrichment of FGFR1 and its downstream AKT signaling are responsible for acquired sorafenib resistance [102]. Although FGF19 amplification may indicate a complete response to sorafenib as discussed above, the FGF19/FGFR4 axis contributes to sorafenib resistance. The depletion of both FGF19 and FGFR4 increases the sensitivity of tumor cells to sorafenib, resulting in enhanced apoptosis and decreased viability. Ponatinib, which is a third-generation multitarget kinase inhibitor, can block FGF19/FGFR4 signaling and reverse sorafenib sensitivity [103]. Hatlen et al. [104] discovered mutations in the gatekeeper and hinge-1 residues on FGFR4, which induced resistance to BLU-554 directly since it destroys the specific covalent interaction sites between BLU-554 and FGFR4. These discoveries allow us to better understand the mechanism underlying acquired drug resistance and show promise for further personalized and targeted therapies.

## 9. Targeting FGF/FGFR

The development of drugs targeting FGFR has involved multiple target kinase inhibitors, pan-FGFR inhibitors, irreversible inhibitors, and reversible inhibitors. Additionally, an increasing number of medications with novel toxicity profiles are appearing. This chapter summarizes the progress of FGFR monotherapy and combination intervention.

### 9.1. Multitarget Kinase Inhibitors

Multitarget kinase inhibitors target a series of growth factor receptors that share a high structural similarity in their tyrosine kinase domains. Lenvatinib, for example, targeting several kinases including VEGFRs, FGFR1-4, PDGFRα, RET, and KIT, has been approved as a frontline treatment alternative to sorafenib for patients with advanced-stage HCC [105,106]. Compared to sorafenib, lenvatinib has more potent activity against VEGF receptors and the FGFR family [1]. The effect of lenvatinib was not inferior to that of sorafenib on the overall survival of patients with unresectable HCC and showed significant benefits in all observed secondary endpoints, including progression-free survival (PFS), the objective response rate (ORR), and time for progression (TTP) [106]. A 2019 meta-analysis concluded that lenvatinib was more favorable for HBV-positive HCC patients than sorafenib, which was immensely valuable since HBV infection is one of the primary etiologies of HCC [107]. Higher serum FGF19 levels were associated with a better response to lenvatinib. The rate of FGF19 baseline in the objective response (OR) and non-OR groups was 2.09 vs. 1.32 at four weeks and 2.19 vs. 1.40 at eight weeks, respectively, showing the clinical response predictive value of circulating FGF19 in early HCC patients [108].

Regorafenib is an FDA-approved oral multikinase inhibitor. The main targets of regorafenib include VEGFR1-3, PDGFR-β, FGFR1, KIT, RET, and B-RAF. Regorafenib prolonged the median survival time of HCC patients to 10.6 months (7.8 months for a placebo) and significantly reduced the risk of death of sorafenib-resistance patients [109]. Regorafenib and cabozantinib exhibited the best PFS and OS benefits and are preferable in refractory HCC patients compared to other agents [110]. These drugs have now been applied in second-line treatment to help HCC patients who show failure after first-line intervention. Considering the promising effects of kinase inhibitors and immune blockade in various cancers, the combination of nivolumab and regorafenib in HCC is being investigated in a clinical trial (ClinicalTrials.gov Identifier: NCT04170556).

TSU-68(SU6668, also known as orantinib) is an oral multiple RTK inhibitor that targets VEGFR2, PDGFR-β, and FGFR, with endothelial cells as its primary target. In phase I/II trial, 8.6% of patients with advanced HCC showed responses to TSU-68, and the median overall survival of all patients was 13.1 months. TSU-68 showed preferable clinical benefits and satisfying safety profiles for patients with HCC, including those who have impaired liver function [111]. Later, a phase II trial of TSU-68 combined with TACE was conducted, but there were no statistically significant effects [112].

There are many other multitarget kinase inhibitors approved for various cancers, such as ponatinib for chronic myeloid leukemia [113], nintedanib for progressive fibrosing interstitial lung diseases [98], and idiopathic pulmonary fibrosis [114,115]. Although multikinase inhibitors may benefit patients with complementary function among different pathways, the on-target toxicity profile of VEGFR is a barrier for long-term treatment. Therefore, an increasing number of selective inhibitors are needed to improve HCC therapy.

### 9.2. Pan-FGFR Inhibitors

JNJ-42756493 (also known as erdafitinib) was approved in the USA in 2019 to treat urothelial carcinoma with specific FGFR2 or FGFR3 alterations [116]. The ORR of JNJ-42756493 is 42% (consisting of 3% CR and 39% PR). Serious adverse effects, including eye disorders (10%), were observed in 41% of patients. Additionally, 68% and 53% of patients experienced dosage interruptions and dose reductions, respectively [117]. A phase I/II clinical trial of JNJ-42756493 in advanced HCC has been conducted, but the data have not yet been published (ClinicalTrials.gov Identifier: NCT02421185).

AZD4547 is an oral selective small molecular inhibitor of FGFR1-3 (IC50 = 0.2, 2.5, and 1.8 nmol/L) but showed significant weaker effects on FGFR4 (IC50 = 165 nmol/L) and much lower selectivity towards VEGFR, IGF2, CDK2, and p38, showing good selectivity among these kinomes [118]. AZD4547 has antitumor effects by inhibiting receptor phosphorylation and the following FRS2, PLCγ, and MAPK pathways, but does not influence kinase insert domain receptor (KDR)-driven carcinogenesis [118]. Multiple clinical trials of AZD4547 are ongoing to test its safety, efficiency, and biomarker value in solid tumors, lymphomas, or myelomas, such as the study of AZD4547 in patients with FGFR1- or FGFR2-amplified tumors or tumors with other genetic changes (ClinicalTrials.gov Identifier: NCT01795768, NCT04439240).

BGJ398 (also known as infigratinib) is widely studied in advanced urothelial carcinoma and intrahepatic cholangiocarcinoma (ICC). The response rate of BGJ398 was 25.4% for patients with advanced urothelial carcinoma and 14.8% for those with ICC, which was favorable compared with those of other novel therapies [119,120]. Patients with FGFR3 mutants were more likely to respond to BGJ398, which may partly be explained by the high specificity of BGJ398 for FGFR3. In advanced urothelial carcinoma, the most frequent AE (advance effect) was hyperphosphatemia (46.3%) [119]. For patients with ICC, the most frequent AE was hyperphosphatemia, followed by fatigue and stomatitis [120]. In general, these AEs were manageable and reversible. There have been no statistics on BGJ398, AZD4547, or other pan-FGFR inhibitor treatments for HCC in clinical trials, which are needed. BGJ398 also showed strong antitumor activity in an HCC patient-derived tumor xenograft (PDX) model expressing high FGFR2 and FGFR3 with acquired sorafenib resistance. This treatment blocked the FGF/FGFR pathway, interfered with the cell cycle and tumor migration, and promoted vascular normalization. However, further clinical trials are required to verify its efficacy and safety in HCC [121].

It is puzzling that most pan-FGFR inhibitors have strong affinities for FGFR1-3 while sparing FGFR4. Simultaneously, most of the adverse effects of pan-inhibitors can be due to FGFR1-3 on-target dose-dependent toxicities. For example, hyperphosphatemia is a typical adverse event of FGFR1-3 inhibition. Patients treated with this pan-FGFR need to be under strict supervision for the blood phosphate concentration and receive intervention measures if necessary. Additionally, a low specificity for FGFR4 makes it ineffective on FGFR4-driven tumors, which has been discussed above as the most potent oncogenic signaling in HCC. FGFR4-selected inhibitors have thus been discovered to relieve the negative reactions and strengthen the blockage of FGFR4-induced HCC.

### 9.3. FGFR4 Inhibitors

As discussed above, FGF4 shares a high sequence identity with FGFR1-3. The discovery of the poorly conversed cysteine at position 552 (Cys552) suggests one of the feasible ways to develop selective inhibitors of FGFR4 [122]. Cys552 is located in the mid-hinge region of the ATP binding site of FGFR4, while it is tyrosine in FGFR1-3 in the same area. Cysteine replacement by tyrosine at this position makes a difference in the ATP-binding pocket size and covalent interaction properties [122]. Based on the character of Cys552 in FGFR4, BLU-9921, BLU-554, H3b-6527, and FGF401 are disclosed one after another.

#### 9.3.1. Irreversible Inhibitors

BLU-9931 is the first selective small molecule for FGFR4. By covalently binding with Cys552 of FGFR4 selectively and irreversibly, BLU-9921 effectively inhibited the enzyme activity of FGFR4s (IC50 = 3 nmol/L) while weakly inhibiting that of FGFR1-3 (IC50 = 591, 493, and 150 nmol/L) [123]. BLU-554 (also known as fisogatinib) is the optimized product of BLU-9931. This molecule is still being investigated in a phase I study for the treatment of advanced HCC. The phase I trial of BLU-554 noted that most of the AEs of BLU-9931 are manageable and acceptable with a maximum tolerated dose (MTD) defined as 600 mg once daily for advanced HCC patients. The median duration of the response was 5.3 months (95% CI, 3.7-not reached) (ClinicalTrials.gov Identifier: NCT02508467) [65]. The concentration of serum FGF19 was also associated with the BLU-554 response. The overall response rate of BLU-554 in FGF19-positive HCC was 17% (11 of 66 patients) and 0% (0 of 32 patients) for FGF19-negative HCC [124].

H3B-6527 also showed a difference between FGFR1-3 and FGFR4, and the IC50 values differ by a thousand times (IC50 values of 0.32, 1, 29, and 1.0 and 1.06 nM). A phase I trial of H3B-6527 in advanced HCC is now recruiting (ClinicalTrials.gov Identifier: NCT02834780) [125].

Although irreversible inhibitors have highly selective characteristics, they may not achieve optimal effects due to the high synthesis rate of FGFR4 [122]. Irreversible covalent binding may have side effects by forming covalent compounds with off-target proteins, with or without hyperreactive cysteines [126]. Furthermore, recent studies have noted that the resynthesis rate of FGFR4 is high [122]. Therefore, a complete and continuous drug intervention is essential to achieving maximum antitumor efficacy. Irreversible inhibitor side effects and the need for sustained drug function are pushing investigators to identify inhibitors that can show ongoing effects during anti-FGFR4 therapy.

#### 9.3.2. Reversible Inhibitors

FGF401 (also known as roblitinib) is a reversible covalent inhibitor, displaying more than a 100-fold selectivity for FGFR4 compared to FGFR1-3. This drug has an increased target residence time of 4.5 h [122]. FGF401 exhibits potent antitumor activity in HCC with aberrant FGF19 overexpression [127]. Novartis pharmaceuticals sponsored a clinical trial of FGF401 from 2014 to 2020. This study aimed to test the maximum tolerated dose and/or recommended phase II dose and efficacy of FGF401 as a monotherapy or in combination with PDR001 in HCC patients positive for FGFR4 and KLB (ClinicalTrials.gov Identifier: NCT02325739). This study has been completed but conclusive data have not been published, and the phase II part of the FGF401 + PDR001 combination was halted due to commercial reasons. FGF401 also performed well in HCC with an FGFR mutation at the gate-keeper (V550M and V550L) and hinge-1 (C552) residues, which is the main reason for acquired BLU-554 resistance [103]. This drug showed no difference in the binding manner between FGFR4^WT^, FGFR4^V550M^, and FGFR4 ^V550L^ and possessed comparable inhibitory effects for both wild type and FGFR4V^550M^ and FGFR4 ^V550L^ mutants (IC50: 6 nM for FGFR4^WT^, 13 nM for FGFR4^V550M^, and 9 nM for FGFR4^V550L^) [126]. These observations indicate that FGF401 may be able to overcome drug resistance.

### 9.4. Combined Therapy

Immune checkpoint inhibitor (ICI) therapy, especially programmed cell death protein 1 (PD-1) and programmed cell death ligand 1 (PD-L1), has shown strong efficacy in the treatment of malignant diseases and is gradually becoming the mainstream in antitumor therapy. The combination of the anti-PD-1 antibody and an antiangiogenic agent has shown potent synergistic effects, which may be due to the close correlation and dependence between tumor angiogenesis and microenvironment suppression [128]. The formation of leaky nascent cells inhibits the infiltration of T cells and the maturation of dendritic cells (DCs) while recruiting Tregs and myeloid-derived suppressor cells (MDSCs) in the microenvironment [128]. Hypoxia also induces the transformation of tumor-associated macrophages (TAMs) from an immune stimulatory M1-like phenotype towards an immune inhibitory M2-like phenotype. This immunosuppressive tumor microenvironment can be reprogrammed by the application of antiangiogenic agents, such as VEGF [129]. In turn, interferon-γ (IFN-γ) secreted from activated T cells can promote the normalization of angiogenesis. Multiple preclinical studies have demonstrated the efficacy of such a combination [130]. There are several associated trials currently recruiting (Table 2). It is highly anticipated that new strategies can improve the survival and response rates of HCC in the future.

## 10. Conclusions and Future Perspectives

Liver cancer was the sixth most frequently diagnosed malignancy and the fourth leading cause of cancer death worldwide in 2018, with hundreds of thousands of people dying annually [113]. HCC accounts for 75–85% of primary liver cancer and is the most common type. Thus far FGF/FGFR signaling has shown promising outcomes in HCC intervention. However, inhibitors targeting FGF/FGFR have not completely solved the challenge faced by other agents, such as acquired resistance and patients’ selective response. The addition of diverse disciplines would improve the efficacy in targeting FGF/FGFR in HCC, such as structural biology. Specific binding dependent on the molecular structures of FGFs/FGFRs is essential for the activation of FGF signaling. Commonly, individual FGFs can bind to multiple FGFRs with differential binding affinities, activating several downstream signaling pathways to varying degrees. These signaling pathways affect the occurrence and development of HCC in diverse ways. Advances in structural biology have enabled us to gain more information about the structures of FGFs/FGFRs and the key factors responsible for their specific binding and subsequent signaling pathway transduction. These findings provide a potential strategy for precisely targeting FGFs/FGFRs to suppress their roles in HCC from a structural perspective.

Management of the administration of nonselective TKIs, selective inhibitors of pan-FGFR to FGFR4 inhibitors, and even reversible covalent FGFR4 inhibitors is important. Given that irreversible or reversible-covalent inhibitors are still in early clinical trials, their safety remains unclear. To address this, researchers need to identify more inhibitors based on a deeper understanding of the FGFR-specific target effects and toxicity profiles and the mechanism of carcinogenicity and resistance. The combined application of FGFR and other drugs is also the next research direction, in relation to kinase inhibitors, immunotherapy, checkpoint inhibitors (CPIs), and antivascular therapy. Together with these interventions, cotargeting the FGFR pathway may overcome the emergence of inevitable resistance mechanisms with single-targeted agents and provide more durable responses to treatment. Targeting FGFR is a promising strategy in the treatment of HCC patients.

## Figures and Tables

**Figure 1 cancers-13-01360-f001:**
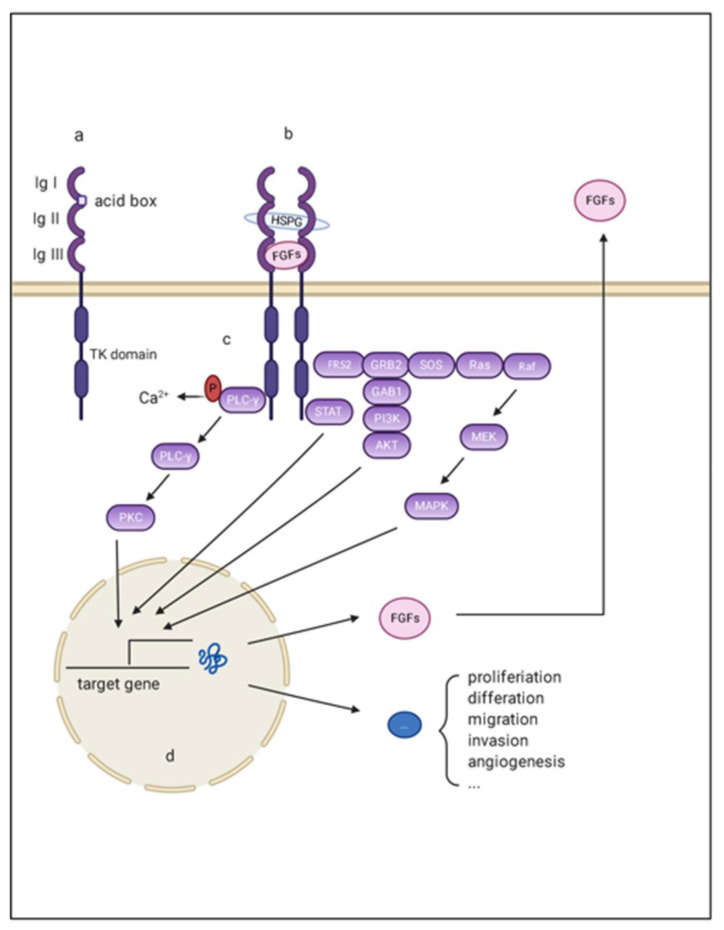
Schematic representation of aberrant FGF/FGFR signaling in HCC. (**a**) FGFR monomer structure: FGFRs are comprised of the extracellular domains linked to intracellular catalytic domains via a single pass transmembrane domain. The extra extracellular domains contains three loops (namely Ig I, Ig II, and Ig III), and an acid box with rich serine; (**b**) The complex composed of FGF, FGFR, and the co-factors: Ig I and the acid box is re-sponsible for signal autoinhibition, while Ig II and Ig III are essential for signal transmission through binding with FGF and the co-factors; (**c**) The intracellular downstream signaling of FGF/FGFR signaling: There are mainly four pathways acting as canonical downstream signaling pathways of FGF-FGFR signal: mitogen-activated protein kinase (MAPK); phosphatidylinositol 3-kinase (PI3-kinase), phospholipase Cγ (PLCγ), and signal transducer and activator of transcription (STAT); (**d**) The target effects of FGF/FGFR signaling: the final effects of these activating downstream pathways are transcriptionally activating a series of target genes that are responsible for multiple hallmarks of HCC.

**Figure 2 cancers-13-01360-f002:**
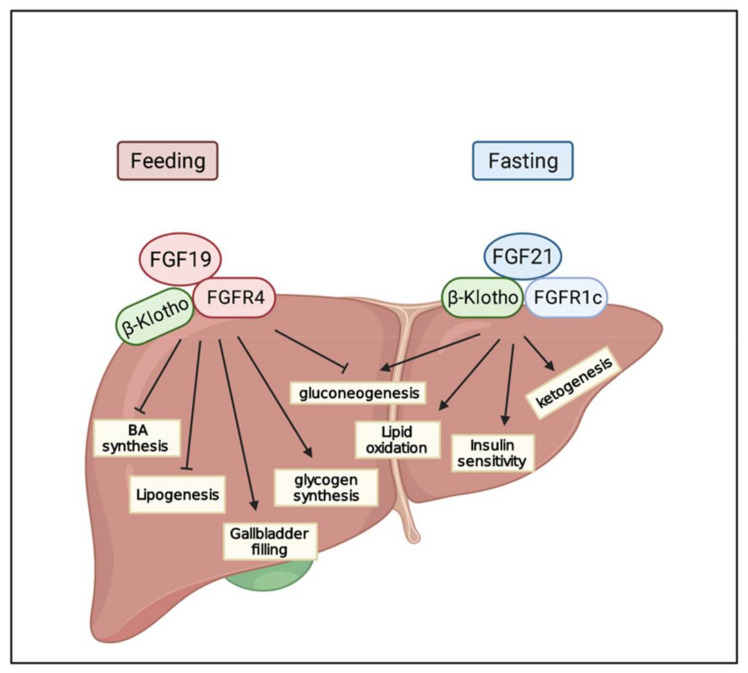
The regulation of FGF19 and FGF21 on liver metabolism. FGF19 is a feeding-response hormone and stimulated followed by increasing bile acid and FXR activation. In turn, FGF19 suppresses the expression of CYP7A1 to inhibit bile acid synthesis. Other than that, FGF19 exerts metabolism roles like insulin, such as stimulating glycogen and protein synthesis and inhibiting gluconeogenesis. FGF19 mainly binds to FGFR4 and β-klotho in the liver. On the contrary, FGF21 is a fasting-induced hepatokine and is partly comparable to glucagon, inducing a transformation to catabolic metabolism through binding to βKlotho-FGFR1c.

**Table 1 cancers-13-01360-t001:** The classification of FGF ligands and their corresponding FGFRs.

FGF Subfamily	FGF	FGFR
FGF1 (paracrine)	FGF1 (aFGF)	All FGFRs
	FGF2 (bFGF)	FGFR1c, FGFR2c, FGFR3-IIIc, FGFR1b, FGFR4
FGF4 (paracrine)	FGF4	FGFR1c, FGFR2c, FGFR3c
	FGF5	FGFR1c, FGFR2 c
	FGF6	FGFR2b
FGF7 (paracrine)	FGF3	FGFR1b, FGFR2b
	FGF7(KGF)	FGFR2b
	FGF10	FGFR2b
	FGF22	FGFR1b, FGFR2b
FGF8 (paracrine)	FGF8	FGFR2c, FGFR3c, FGFR4
	FGF17	FGFR2c, FGFR3c, FGFR4
	FGF18	FGFR2c, FGFR3c, FGFR4
FGF9 (paracrine)	FGF9	FGFR3b, FGFR3c
	FGF16	FGFR2c, FGFR3c, and FGFR4
	FGF20	FGFR1c
FGF19 (endocrine)	FGF15/19	FGFR4, FGFR1c, FGFR2c, FGFR3c
	FGF21	FGFR1c, FGFR3c
	FGF23	FGFR1c, FGFR2c, FGFR3c, FGFR4
FGF11 (Intracrine)	FGF11	
	FGF12	
	FGF13	
	FGF14	

**Table 2 cancers-13-01360-t002:** Ongoing trials of FGF/FGFR targeted therapies for HCC.

Drug	Drug Target	Conditions	Status	Phase	NCT Number
Regorafenib	VEGFR1–3, PDGFR, RAF kinase, FGFR1–2	HCC	Not recruiting	Phase 2	NCT04476329
BLU554	FGFR4	HCC	Active, not recruiting	Active, not recruiting	NCT02508467
H3B-6527	FGFR4	HCC	Recruiting	Phase 1	NCT02834780
Regorafenib + Nivolumab	VEGFR1–3, PDGFR, RAF kinase, FGFR1–2 + PD-1	HCC	Recruiting	Phase 1 Phase 2	NCT04170556
Pembrolizumab + Lenvatinib	PD-1 + VEGFR1–3, PDGFR, FGFR1–4, RET	Liver Transplant Complications; HCC Recurrent	Recruiting	Not Applicable	NCT04425226
Durvalumab + Lenvatinib	PD-L1 + VEGFR1–3, PDGFR, FGFR1–4, RET	Liver carcinoma; Liver Transplant; Complications	Recruiting	Not Applicable	NCT04443322
Camrelizumab + Lenvatinib	PD-1 + Multitarget kinase inhibitors	HCC	Recruiting	Phase 1 Phase 2	NCT04443309
Lenvatinib + Toripalimab	VEGFR1–3, PDGFR, FGFR1–4, RET + PD-1	HCC	Recruiting	Phase 2	NCT04368078
Lenvatinib + TACE versus Sorafenib + TACE	VEGFR1–3, PDGFR, FGFR1–4, RET	HCC; Tumor Thrombus	Enrolling	Phase 4	NCT04127396

## Data Availability

Data sharing not applicable. No new data were created or analyzed in this study.

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
