# Peer review of "FGF/FGFR Signaling in Hepatocellular Carcinoma: From Carcinogenesis to Recent Therapeutic Intervention"

_cancers, 2021, doi:10.3390/cancers13061360_

Round 1

Reviewer 1 Report

This is a review article focused on FGF/FGFR in HCC. It is very extensive and deals with a timely topic. However, it is very difficult to read due to English language issues and typographical errors. In addition, the figures are confusing and the figure legends are too laconic so both aspects require a great deal of improvement. By contrast, the sole table is too simplistic and should be expanded to provide more information about each listed study and perhaps  also expanded to include some additional studies. It will take a fair amount of effort to fix all these deficiencies, but it is doable. 

Author Response

Thanks very much for taking your time to review this manuscript. I really appreciate all your comments and suggestions. According to your suggestions, we have made the following revisions on this manuscript:  

1. We have polished this article and correct the grammatical problems.

2. We have revised Figure 1 with more legends and less unnecessary contents to make it more precise and readable.

3. We have redrawn Figure 2 to introduce the information about FGF metabolism instead of FGF angiogenesis.

4. In table 2, we have made some adjustments to provide contents that is more necessary and readable.

Reviewer 2 Report

Comprehensive review.

No major comments to section 2-8. Along with the company-given trial name (i.e: BGJ398), all drugs should be identified also with their current name (BGJ398 is Infigratinib, FGF401 is Roblitib, BLU554 is Fisogatinib, etc...)

Section 9 should be revised and updated. It is singular the lack of referring to the use of pemigatinib (INCB054828) in cholangiocarcinoma (Lancet Oncol 2020). In addition, data from the NCT02325739 trial on Roblitinib in HCC have been disclosed by Novartis and should be reported. Data from reference Hepatology 2019; 69: 943-958 on Infigratinib in HCC should be reported and discussed.

Author Response

Thank you for reviewing our manuscript and offering valuable advice. In accordance with your suggestions, we have made the following revisions to our manuscript: 

1. We have added the current names of each drugs in section 9.

2. We have also added the information about Infigratinib (9.2) and Roblitinib (9.3.2).

3. Considering that this article is mainly focused on HCC, we do not involve drugs in cholangiocarcinoma. If you think this part is indispensable, we can add it later.

Thank you very much for your reply !

Reviewer 3 Report

This is a comprehensive and detailed review about the aberrant FGF/FGFR signaling in HCC. However I have 2 major points about this review

1. Authors need to cite the most recent general review for HCC by Llovet JM et al, https://doi.org/10.1038/s41572-020-00240-3

2. It is known that  recurrent focal chromosome amplifications
in CCND1, FGF19, VEGFA, MYC or MET leading
to over- expression result in the activation of various
oncogenic signalling pathways, including of receptor
tyrosine kinases. Although cancer driver gene mutations
accumulate randomly, specific genes are related to
precise molecular HCC subclasses, defined by transcriptomic
profiles and histological phenotypes. 

Please see papers by

Calderaro J. et al. Journal of Hepatology 2017 :67 :727–738

Calderaro, J., Ziol, M., Paradis, V. & Zucman- Rossi, J.
Molecular and histological correlations in liver cancer.
J. Hepatol. 71, 616–630 (2019).

In all these 3 landmark papers, it is described the Clinical, pathological and molecular features of macrotrabecular-massive HCC ,with FGF19/CCND1 amplification (G3 subclass,Boyault G1–G6). This is absent from this manuscript ,and has to be added properly, and connected with the general discussion of FGF/FGFR signaling.

Minor points: 1.The quality of both figures is suboptimal.

2. Table 1. Error ...parecrine

Figure 1.  Error....aotucrine

Author Response

We really appreciate you for your carefulness and conscientiousness. Your suggestions are really valuable and helpful for revising and improving our paper. According to your suggestions, we have made the following revisions on this manuscript:  

1. We have studied all of the reference involved in your revision and cited  most of these references in our review.

2. We have added information about macrotrabecular-massive HCC and connect it with our review.

3. We have revised Figure 1 with more legends and less unnecessary content to make it more precise and readable.

4. We have redrawn Figure 2 to introduce the information about FGF metabolism instead of FGF angiogenesis.

5. In table 2, we have made some adjustments to provide contents that are more necessary and readable.   Thank you very much for your reply!

Round 2

Reviewer 1 Report

The authors have addressed some of the previous concerns with improved figure legends and greater attention to English language issues. However, there is still the need to extensively re-write the paper by a person proficient in English.

For example:

I believe the word “circulation” should replace “cycle” and the word “actions“ is required at the end of this sentence: “FGFs with low affinity to HSPG (FGF19, FGF21, FGF23) enter the cycle and exert their functions in the form of endocrine [16]”

Corrected sentence: “FGFs with low affinity to HSPG (FGF19, FGF21, FGF23) enter the circulation and exert their functions in the form of endocrine actions [16]”

Author Response

Thank you for your suggestion! Considering our unsatisfactory English proficiency, we have purchased the English editing services from MDPI (English-27290). In this revision, we have carefully revised our manuscript and check some grammatical problems. We hope to meet your approval, thank you!

Reviewer 2 Report

The work has significantly improved.

Author Response

Thank you for your suggestion and approval! In this revision, we have carefully revised our manuscript and check some grammatical problems.

Reviewer 3 Report

My points to the authors are all satisfied in the revised version

Author Response

(The authors gave the same response as above.)

Round 3

Reviewer 1 Report

The revised version is improved. At this juncture there is no need to add additional material or references. However, there are still numerous errors due to English language issues. Here are 10 examples that were easy to correct, but there many more:

L20-21:several studies have emphasized the development….

L45: meaning that HCC can only be detected…

L50: High light: highlight

L102-103: Klotho proteins confer stability and preferential binding of endocrine FGFs to their respective FGFRs.

L112: FGF19

L204: with KLB stabilizing their….

L212: an average of 53% of mice….

L219: more fragile to CCL4 exposure

L250-251: downstream molecular target of miR-188-5p….

L285: Aberrant

There are also many sentences that are still very confusing and not easy to understand.

In addition, there are numerous typographical errors, and some of these can be crucially important. For example, on line 626, “the ATP binding site of FGF4” should be the ATP binding site of FGFR4.

The authors need to seek the help of someone who is proficient in English, and to very carefully proof the paper, word by word. They should not add additional material since they are thereby introducing no English language errors and the review is in danger of becoming too lengthy.

Author Response

Thank you for your suggestions. We have carefully checked  and polished this manuscript in a professional websites. Hope to meet your approval.